# Virtual Reality Exercise Program Effects on Body Mass Index, Depression, Exercise Fun and Exercise Immersion in Overweight Middle-Aged Women: A Randomized Controlled Trial

**DOI:** 10.3390/ijerph20020900

**Published:** 2023-01-04

**Authors:** Eun-Young Seo, Yeon-Suk Kim, Yu-Jin Lee, Myung-Haeng Hur

**Affiliations:** 1Department of Nursing, Gyeongbuk College of Health, Gimcheon 39525, Republic of Korea; 2Department of Nursing, Chung Cheong University, Cheongju 28171, Republic of Korea; 3College of Nursing, Eulji University, Uijeongbu 11759, Republic of Korea

**Keywords:** virtual reality, exercise, overweight, body mass index, depression, immersion

## Abstract

Background: This study explored the effects of a virtual reality exercise program on overweight middle-aged women. Methods: This randomized controlled trial included women 40–65 years of age with a body mass index (BMI) of 23 kg/m^2^ or more living in Daejeon City. The virtual reality environment was set up by attaching an IoT sensor to an indoor bicycle and linking it with a smartphone, enabling exercise in an immersive virtual reality through a head-mounted display. Results: In the virtual reality exercise group, the BMI was significantly decreased after the 8-week intervention compared with the baseline value (F = 59.491, *p* < 0.001). The depression scores were significantly different among the three groups, with the intervention effect being more significant in the virtual reality exercise group than in the indoor bicycle exercise and control groups (F = 3.462, *p* < 0.001). Furthermore, the levels of exercise fun (F = 12.373, *p* < 0.001) and exercise immersion (F = 14.629, *p* < 0.001) were significantly higher in the virtual reality exercise group than in the indoor bicycle exercise and control groups. Conclusions: The virtual reality exercise program positively affected the BMI and the levels of depression, exercise fun, and exercise immersion in overweight middle-aged women. It is an effective home exercise program for obesity management in this population.

## 1. Introduction

While rapid economic growth and urbanization have enhanced the convenience of modern life, the resulting decrease in physical activity and lack of exercise have led to an increased obesity rate worldwide. More than 1 billion adults are overweight globally, with at least 300 million being clinically diagnosed with obesity [1]. Obesity increases the risk of various diseases, including hypertension, diabetes, arteriosclerosis, hyperlipidemia, cardiovascular disease, and certain cancers. With an increased risk of premature death by obesity, it is no longer a cosmetic problem but a disease threatening the welfare of people worldwide [2]. In South Korea, the Korea National Health and Nutrition Examination Survey data reported an obesity prevalence of 34.6% among adults in 2018, demonstrating a steady increase. These data also showed that the prevalence of obesity decreased with increasing age in men after age 40. In contrast, it increased with increasing age in women after age 40, suggesting the need for obesity management in midlife [3].

The reason for middle-aged women having a higher obesity rate than men of the same age is that women accumulate excessive body fat during menopause due to decreased metabolism and the decreased secretion of growth hormones and estrogen, which are lipolytic hormones [4]. Compared with those with a normal weight, middle-aged women who are overweight or have obesity have a higher risk for cancer, such as breast and ovarian cancer, and are more vulnerable to chronic diseases, such as diabetes and hypertension [5]. Obesity in women has also been found to affect the health-related quality of life due to depression and activity restriction [6]. In a prospective cohort study of women aged ≥50 years, obesity in middle-aged women was confirmed to affect the long-term risk of developing depression [7]. Obesity can be said to be a negative consequence of endocrine diseases, neurological abnormalities, genetic influences, and environmental causes, including eating habits and a lack of exercise, that cause imbalances in intake and consumption [8]. To overcome obesity, consistent efforts are required, including diet, exercise, and behavioral therapy for stress reduction [9]. Among them, exercise positively affects the immune function and obesity factors in middle-aged women, helps weight loss, prevents metabolic diseases, and is important in terms of delaying aging and preventing diseases [10].

However, various factors are decreasing the exercise rate, one being the coronavirus disease 2019 (COVID-19) pandemic. In South Korea, after the first case of COVID-19 was confirmed in January 2020, the country began implementing social distancing by announcing guidelines on infection control, prevention, and behavior [11]. Social distancing is an effective strategy for reducing the spread of infectious diseases by minimizing contact between individuals and groups; however, continuous social isolation also causes various side effects, such as depression, anxiety, panic disorder, and decreased physical activity [12]. Another cause of the reduced exercise rate is fine dust, an environmental factor. Fine dust contains various components, such as metal, ionic, and carbon [13]. As the concentration of fine dust has gradually increased over the years, the social interest in its effects on the human body is also increasing. The effects of fine dust on human health vary depending on the particulate matter size, number, and exposure time [14]. Prolonged exposure to particulate matter lowers immunity, causing acute inflammation, likely leading to respiratory diseases such as asthma and pneumonia, cardiovascular diseases, and various cancers. It also increases respiratory disease hospitalization and mortality rates [15]. The decrease in outdoor activities for these reasons is leading to a decrease in the exercise practice rate. Experts predict that another infectious disease may appear after the COVID-19 epidemic and that our daily lives must be prepared for such environmental changes [16]. One aspect of such change is a shift from traditional physical activity performed outdoors or in groups to indoor exercise, particularly at home [17]. Among the various types of home exercise, including treadmill exercise, indoor cycling, yoga, and others, indoor cycling is a simple aerobic activity that can be performed at home with no spatial or temporal restrictions, being suitable for busy people in the modern era. It is also suitable for overweight people, allowing them to exercise without straining their joints [18]. However, it is important to remain interested until obtaining the desired amount of exercise, and many people become bored of repetitive routines and stop participating in exercise.

Recently, virtual reality indoor exercise programs have been developed to enhance the engagement and interest in indoor exercise to compensate for these shortcomings [19]. Virtual reality is a core novel technology that will lead the fourth industrial revolution; this cutting-edge technology allows users to obtain real-world experiences by interacting in a virtual world provided by computers, similar to reality [20]. The advantage of virtual reality is that it allows one to experience the computer-generated world as if it were real, creating a sense of “being there” in users’ minds and allowing them to experience immersion [21]. In addition, the level of participation is improved by improving cognition and concentration and by the pleasure of virtual reality [22]. Virtual reality-applied exercise does not have any time and space restrictions, unlike exercise in the real world, and it improves enjoyment and energy and reduces fatigue [23,24]. The fun factor of virtual reality can effectively promote immersion and enhance motivation and learning [25]. Recently, virtual reality exercise programs have been applied in various fields of the medical sector, with positive results, such as improvements in physical function and exercise performance among older adults, enhanced cognitive function in patients with dementia and stroke, and fall prevention in patients with Parkinson’s disease [26,27,28,29].

Virtual reality is applicable in various fields in the post-COVID-19 era. A new paradigm shift is required in exercise intervention methods with the change to non-face-to-face life patterns [30]. To date, no studies have investigated the application of virtual reality programs among the general public in South Korea. Therefore, in this study, we developed a new exercise intervention method that provides a virtual reality indoor cycling experience using a head-mounted display (HMD). We compared its effects with regular indoor bicycle exercise and no exercise.

In this study, the effect of the exercise intervention method applying virtual reality on the body mass index (BMI), depression, exercise fun, and exercise immersion in middle-aged women was investigated.

**Hypothesis** **1.**
*There will be a difference in the BMI of the virtual reality exercise, indoor bicycle exercise, and control groups to which the virtual reality exercise program is applied.*


**Hypothesis** **2.**
*There will be a difference in depression between the virtual reality exercise, indoor bicycle exercise, and control groups to which the virtual reality exercise program is applied.*


**Hypothesis** **3.**
*There will be a difference in the exercise fun between the virtual reality exercise, indoor bicycle exercise, and control groups to which the virtual reality exercise program is applied.*


**Hypothesis** **4.**
*There will be differences in exercise immersion among the virtual reality exercise, indoor bicycle exercise, and control groups to which the virtual reality exercise program is applied.*


## 2. Materials and Methods

### 2.1. Study Design and Population

The data collection of this study was conducted from February to May 2021 for overweight middle-aged women living in Daejeon City through a recruitment notice at Eulji University Hospital. This randomized controlled trial examined the effects of an 8-week virtual reality exercise program on the BMI and levels of depression, exercise fun, and exercise immersion among overweight middle-aged women living in the city and compared to those with indoor bicycle exercise and no exercise (Figure 1). The study population included women aged 40–65 years with a BMI of ≥23 kg/m^2^, those with mild depressive symptoms with a PHQ-9 score of 5–9, those who had not participated in an indoor bicycle exercise program for the past 6 months, those without visual or auditory impairments, those who could use a smartphone, and those who agreed to participate in the study. However, participants with a history of orthopedic diseases or recent surgery, and those who complained of cybersickness and headaches, were excluded.

### 2.2. Randomization

The recruitment order was coded to prevent participant allocation bias, and the participants were randomly assigned to the virtual reality exercise, indoor bicycle exercise, or control groups using the random Excel (version 19.0) function. For the randomized controlled trial study, the study was conducted after registering with the Clinical Research Information Service (CRIS registered number: KCT0006175).

### 2.3. Interventions

The exercise program was constructed in consideration of the exercise type and intensity according to the recommendations of the American College of Sport Medicine, in consultation with an internal medicine specialist, a professor of nursing, and an exercise prescriber. Exercises suitable for overweight people or people with obesity include low-impact aerobic exercises, such as walking, stationary bicycle riding, running, and swimming, which do not strain joints [31]. This study utilized indoor cycling because it does not place significant strain on the joints, is easy to access, and can be performed at home. In terms of exercise intensity, a low exercise intensity was applied, suitable for overweight people or people with obesity [32]. The intensity of the indoor bicycle exercise was set at gear 2 (out of 10 gears), and the perceived exercise intensity was considered [33] by setting the intensity of exercise such that participants became “slightly out of breath”.

The exercise program consisted of warm-up, main, and finishing exercises. Warm-up exercises consisted of warm-up stretching before exercise, which relieved the stimulation of the heart and muscles and improved the exercise capacity by improving blood flow. Indoor bicycle exercise was applied as the main exercise. As a finishing exercise, cool-down stretching was performed to accelerate the decomposition of lactic acid accumulated in the blood and help with fatigue recovery after the main exercise. Since an adequate exercise time for people with obesity is 40–60 min [34], the exercise intervention in this study consisted of 10 min of warm-up, 30 min of the main exercise, and 10 min of finishing exercise, for a total duration of 50 min. As for the frequency of exercise, exercising once a week caused muscle pain and fatigue, exercising twice a week had a slight effect, and exercising more than three times a week had the maximal effect [35]. Therefore, this study performed exercises three to five times a week.

The virtual reality exercise group performed the virtual reality indoor cycling program, and the indoor bicycle exercise group performed the regular indoor cycling program. In contrast, the control group was allowed to perform daily activities for 8 weeks without an exercise intervention. The exercise program in the two exercise groups was the same as described above. The only difference was that a virtual reality environment was provided during the indoor bicycle exercise in the virtual reality exercise group. This was achieved by attaching an IoT sensor to the indoor bicycle, connected to the VR*Fit* application, downloaded on a smartphone. After setting the virtual reality background and music on the VR*Fit* application screen, the set virtual background and music would appear. Exercise would begin when turning the bicycle pedal after attaching the smartphone to the HMD. The two groups were encouraged to exercise three to five times weekly for 8 weeks. The exercise programs were provided at home. The 8-week exercise application process for the virtual reality and indoor bicycle exercises is as follows: during the VR*Fit* application-linked exercise, a stamp was placed on the VR*Fit* record table, and participants were instructed to send it to the researcher at the endpoint after 8 weeks of exercise. Indoor bicycle exercise enabled the transmission of the indoor bicycles’ exercise records as pictures. The warm-up and cool-down stretching exercises were conducted by following the instructions on the handout provided by the author.

### 2.4. Outcome Measurement

Before the start of the intervention, all participants underwent height and weight measurements to calculate their BMI. They also completed questionnaires on general data (marital status, education, employment, income, chronic diseases, alcohol consumption, and medication use), depression symptoms over the past 2 weeks, and the amount of fun and immersion related to daily exercise. The BMI and levels of depression, exercise fun, and exercise immersion were measured in the same way at weeks 4 and 8 from the start of the intervention in all three groups.

#### 2.4.1. BMI

A manual extensometer and a digital scale for weight measurement were used for height measurement (H5, CAS, Seoul, Korea). The BMI was calculated as follows: BMI = weight (kg)/[height (m)]^2^.

#### 2.4.2. Depression

The Korean version of the Patient Health Questionnaire-9 (PHQ-9) was used to measure the level of depression. The PHQ-9 is a depression scale based on the Diagnostic and Statistical Manual of Mental Disorders, 4th Edition. The scale consists of a total of nine items scored on a four-point Likert scale (0–3), with 3 indicating “almost every day” and 0 indicating “never”. The total score is 27 points, with a higher score indicating more severe depressive symptoms. The scale’s reliability was assessed as high, with a Cronbach’s alpha of 0.80, with the same findings in this study (Cronbach’s α = 0.80).

#### 2.4.3. Exercise Fun

A numeric rating scale was used to measure the subjective level of exercise fun. The following four items were rated on a scale from 0 to 10: (1) Exercise is fun; (2) Exercise made me feel good; (3) Exercise relieved stress; and (4) Exercise made me feel happy. The total score ranged from 0 to 40 points, with higher scores indicating a higher level of exercise fun.

#### 2.4.4. Exercise Immersion

To measure exercise immersion, we used the Sports Immersion Scale, developed by Jung [36], through modification of the Expansion of Sport Commitment Model scale developed by Scanlan, Carpenter, Schmidt, and Keeler. The scale consists of 12 items in two cognitive and behavioral immersion areas. Each item was scored on a five-point Likert scale (1–5), with “1” indicating “strongly disagree” and 5 indicating “strongly agree”. The total score ranged from 12 to 60 points, with a higher score indicating a higher level of exercise immersion. The scale’s reliability was assessed as high, with a Cronbach’s alpha of 0.86–0.94, with similar findings in this study (Cronbach’s α = 0.94).

### 2.5. Sample Size Calculation

The sample size for this study was calculated using the G-Power 3.1.9 program. By selecting repeated-measures analysis of variance (ANOVA) and setting the effect size [37] at 0.2, significance level at 0.05, power at 0.9, number of groups at 3, and number of measurements at 3, the obtained number of samples was 69. To account for a 10% dropout rate during the intervention, the final sample size was defined as 75.

### 2.6. Statistical Analysis

All data were analyzed using IBM SPSS Statistics for Windows, version 26.0. The participants’ general characteristics were analyzed by frequency, percentage, and average, and the homogeneity of the general characteristics was analyzed using ANOVA and the χ^2^ test. Prior homogeneity of the dependent variables of the three groups was analyzed using ANOVA. The three groups were compared using ANCOVA to verify the post-intervention effects, and post-hoc analysis was conducted using Bonferroni’s method. Repeated-measures ANOVA was employed to test the effect difference according to the time change. When the sphericity was not satisfied due to the sphericity test, Wilks’ Lambda multivariate verification was performed for analysis. The effect sizes of group and time were analyzed with partial eta squared (partial η²) to explain the degree of influence of the independent variable on the dependent variable. If the partial η^2^ is 0.01, 0.06, or ≥0.14, the effect size is small, medium, or large, respectively. Therefore, the closer the value of partial η^2^ is to 1, the larger the average difference between groups and the smaller the error [38].

## 3. Results

### 3.1. Homogeneity of the Participants’ General Characteristics and Prior Dependent Variables

A total of 75 participants were recruited, with 25 in each group. During the study, two participants in the virtual reality exercise, one in the indoor bicycle exercise, and two in the control group dropped out. Thus, 70 participants completed the study: 23 in the virtual reality exercise, 24 in the indoor bicycle exercise, and 23 in the control group (Figure 2).

There was no significant difference in the general characteristics among the three groups at baseline, indicating homogeneity. Further, the one-way ANOVA for homogeneity of the dependent variables showed no significant differences in the BMI or the level of depression, exercise fun, and exercise immersion, indicating the homogeneity of the three groups (Table 1).

### 3.2. Outcome Variables

The study outcomes in all three groups are presented in Table 2.

#### 3.2.1. BMI

At baseline, the average BMI among all participants was 25.49 kg/m^2^, and the average body weight was 65.08 kg.

In the virtual reality exercise group, the mean BMI decreased from 26.04 kg/m^2^ at baseline to 25.35 kg/m^2^ at week 4 and 24.65 kg/m^2^ at week 8, with a statistically significant difference between the baseline and 8-week values. In the indoor bicycle exercise group, the BMI also decreased from baseline to week 8, but the difference was not statistically significant. In the control group, no changes in BMI were observed over the 8 weeks.

In the comparison among groups, there were significant differences in the BMI among the three groups at week 4 and week 8. Repeated-measures ANOVA showed a significant difference in the group–time interaction (F = 59.491, *p* < *0*.001). The partial eta-squared, the effect size of the virtual reality exercise program according to group and time point, was 0.640.

#### 3.2.2. Depression

The depression scores decreased from baseline to week 8 in all three groups, with the most significant difference in the virtual reality exercise group (5.48 ± 4.50 at baseline vs. 2.48 ± 1.47 at week 8). At baseline, there was no significant difference among the groups; however, the depression scores at week 4 (F = 7.804, *p* < 0.001) and week 8 (F = 9.502, *p* < 0.001) showed a statistically significant difference among the three groups. Repeated-measures ANOVA showed a significant difference in the group–time interaction (F = 3.462, *p* = 0.010). The partial eta squared, the effect size of the virtual reality exercise program according to group and time point, was 0.094.

#### 3.2.3. Exercise Fun

From baseline to week 8, the total exercise fun score increased in the virtual reality exercise group (17.57 ± 8.78 vs. 32.83 ± 4.61) and decreased in the indoor bicycle exercise group (21.42 ± 8.21 vs. 18.92 ± 7.10); no differences were observed in the control group. Furthermore, at baseline, there was no significant difference among the groups; however, the total exercise fun scores at week 4 (F = 33.731, *p* < 0.001) and week 8 (F = 40.554, *p* < 0.001) showed a statistically significant difference among the three groups. Repeated-measures ANOVA showed a significant difference in the group–time interaction (F = 12.373, *p* < 0.001). The partial eta squared, the effect size of the virtual reality exercise program according to group and time point, was 0.273 (Figure 3).

#### 3.2.4. Exercise Immersion

Similar to the exercise fun score, from baseline to week 8, the exercise immersion score increased in the virtual reality exercise group (30.00 ± 7.20 vs. 45.91 ± 5.95) and decreased in the indoor bicycle exercise group (32.54 ± 8.61 vs. 29.13 ± 6.84). No notable differences were observed in the control group. In addition, at baseline, there was no significant difference among the groups; however, the exercise immersion scores at week 4 (F = 39.187, *p* < 0.001) and week 8 (F = 52.346, *p* < 0.001) showed a statistically significant difference among the three groups. Repeated-measures ANOVA showed a significant difference in the group–time interaction (F = 14.629, *p* < 0.001). The partial eta squared, the effect size of the virtual reality exercise program according to group and time point, was 0.307.

## 4. Discussion

The results of this randomized controlled trial showed that the 8-week virtual reality exercise program positively affected the BMI and the levels of depression, exercise fun, and exercise immersion in overweight middle-aged women.

A significant interaction was observed between time and the groups, including a difference in all three groups’ BMI, following the 8-week intervention. After 8 weeks of exercise, the mean BMI and body weight decreased by 1.39 kg/m^2^ and 3.5 kg in the virtual reality exercise group and by 0.38 kg/m^2^ and 0.95 kg in the indoor bicycle exercise group, respectively. In previous studies that applied an exercise intervention for 8 weeks, the weight decreased by 1 kg with a fast walking exercise [39]. The BMI decreased by 0.5 kg/m^2^ and 1.36 kg/m^2^ with rhythmic gymnastics exercise [40] and combined exercise [41], respectively, indicating a significant effect of an 8-week exercise protocol on BMI and body weight. In particular, the post-hoc analysis in this study revealed a significant difference between the virtual reality exercise and indoor bicycle exercise groups. Even with the same indoor bicycle exercise, the exercise intervention using virtual reality was more effective. Virtual reality exercise programs implemented in various age groups, such as college students [42], middle-aged adults [43], and older adults [26,44], exerted a positive effect on calorie consumption, an improvement in physical function, and an improvement in muscle strength and exercise function. Our findings indicate that the virtual reality exercise program used in this study can have a more significant positive effect on body weight and BMI than an indoor cycling exercise alone by increasing the exercise effect. Obesity is not only recognized as a disease, but several studies have confirmed that it is an important cause of chronic disease. In recent years, the number of obese people in many countries around the world, including Korea, has been gradually increasing, and the prevalence of obesity-related chronic diseases is also increasing, so the problem of obesity is emerging. For this reason, it is necessary to establish a systematic management plan in Korea. In conclusion, the virtual reality exercise program was effective as an at-home workout program as the BMI decreased significantly compared to that in the indoor bicycle exercise and control groups after 8 weeks of exercise.

In this study, depression symptoms were measured and analyzed using the Korean version of the PHQ-9. The average depression score at baseline was 5.8, suggesting that the exercise program participants had mild depression. After the 8-week intervention, the number of participants with a PHQ-9 score of 5 or higher decreased from 11 to 1 in the virtual reality exercise group and from 16 to 8 in the indoor bicycle exercise group. Additionally, there was a significant difference in the depression scores among the three groups; compared with the indoor bicycle exercise and control groups, the virtual reality exercise group showed a statistically significant decrease in depression scores after the intervention. Therefore, the exercise intervention using virtual reality was found to be more effective in reducing depression. It is difficult to identify the underlying mechanism by which the virtual reality exercise program improved depressive symptoms as it involves psychological and neurobiological interactions. However, this is a common result, with several studies reporting a more significant effect on depression when virtual reality training was applied to patients with stroke [45], adults 19–50 years of age [46], and older adults [47,48]. A virtual reality exercise program seems to be an effective intervention method for reducing depression. Virtual reality enables users to indirectly experience situations that are not easy to experience. The audiovisual stimuli received through the screen may have aroused interest, with the interaction pleasure having a psychologically positive effect on depression.

Regarding the levels of exercise fun and exercise immersion in this study, the interaction between time and group was significant, indicating a difference in exercise fun and exercise immersion scores among the three groups according to time. The application of an exercise intervention using a virtual reality therapy method was effective as the post-hoc analysis revealed a significant difference between the virtual reality exercise and indoor bicycle exercise groups. As one of the technological fields that will lead the next generation, virtual reality attracts attention as a technology that can provide improved engagement and value [49]. In a three-dimensional virtual space, participants indirectly experience situations that cannot be experienced in the real world, and the experience of realism and immersion engages them [50]. In this study, a virtual reality program provided indirect experiences of various environments for the exercise. The fun factor of the exercise was increased by the sense of interaction with the surrounding environment. In a previous study on exercise immersion, applying virtual reality to a cycling exercise game increased immersion, which motivated participants in the virtual reality group to move a longer distance [51]. In addition, a virtual reality program using a sensor designed to capture physical movements allowed participants to immerse themselves in the virtual reality world easily, and participants could become more actively engaged in the experience [52]. In this study, a sensor capturing participants’ movement was mounted on an indoor bicycle to increase the effectiveness of exercise immersion in the virtual reality world. Wearing the HMD disconnected the participants from the outside world, and they perceived the surrounding environment to be changed by their actions as if they were exercising in the real world, which increased exercise immersion and its effects. The best advantage of the virtual reality exercise program is that the external environment has few limitations and can enhance the user’s sense of immersion. In addition, since the approach is useful, it has a high possibility of generalization and can induce interest and motivation better than monotonous exercise programs, enabling continuous exercise.

There are some limitations to the study. While the duration of the intervention was the same as in the exercise program, the prolonged experimental period made it difficult to control for differences in individual living environments and diets, which may have affected the variables. Therefore, to control the variables of BMI, it is necessary to investigate them more objectively, in future studies, including nutritional records. Furthermore, when applying immersive virtual reality, it was difficult for participants to observe surrounding objects, predisposing them to possible cybersickness and requiring the investigator’s attention. For the virtual reality exercise program to be continuously applied and to promote of the continuation of exercise, it is necessary to further develop various programs.

## 5. Conclusions

The virtual reality exercise program used in this study effectively reduced the BMI and the level of depression, increased the level of exercise fun and exercise immersion, and enhanced the exercise effects in overweight middle-aged women. However, the continuity of these effects requires further verification through repeated studies in the future. Nonetheless, increasing the engagement and immersion in exercise can be used as an intervention to improve the BMI and reduce depression in overweight middle-aged women.

## Figures and Tables

**Figure 1 ijerph-20-00900-f001:**
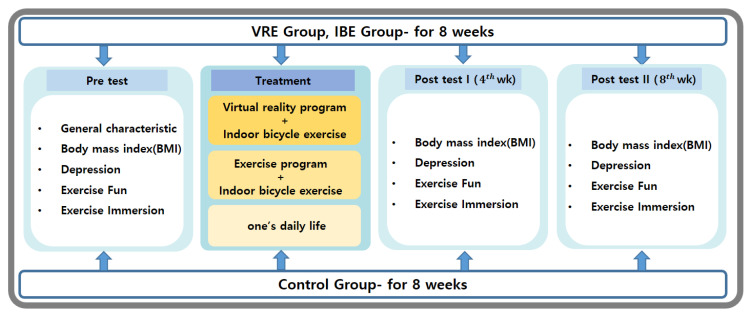
Study design.

**Figure 2 ijerph-20-00900-f002:**
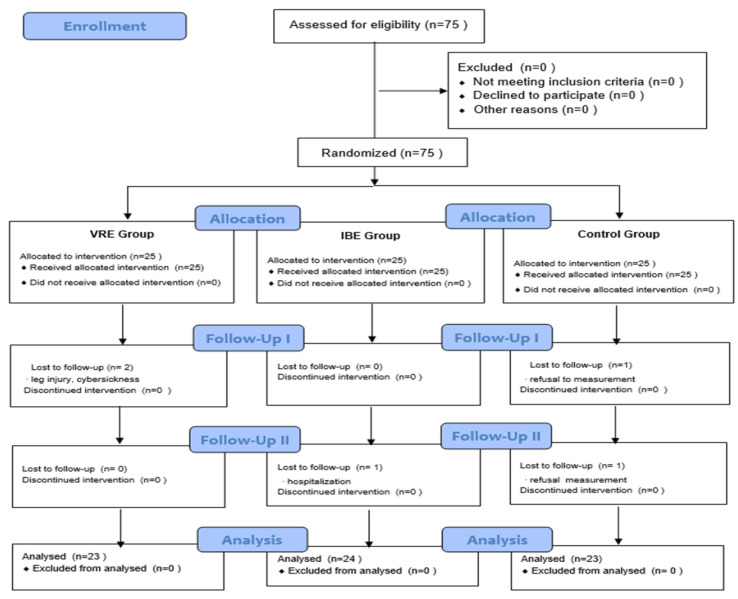
Flow diagram.

**Figure 3 ijerph-20-00900-f003:**
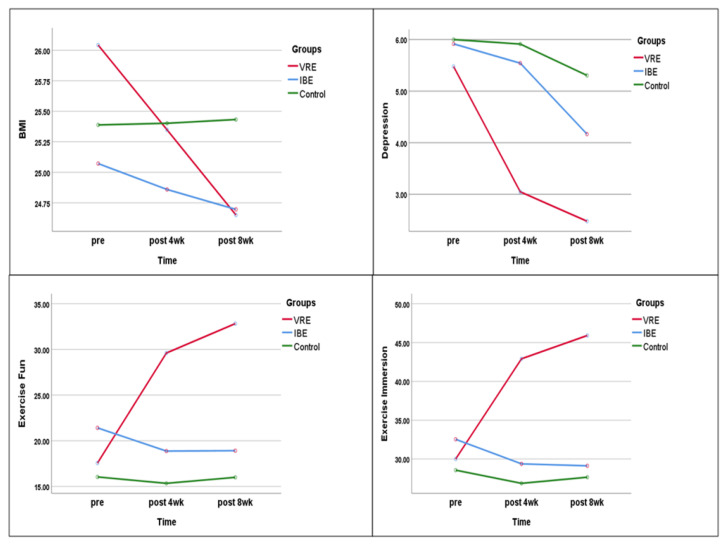
Comparison of BMI, depression, exercise fun, and immersion among groups.

**Table 1 ijerph-20-00900-t001:** Homogeneity of the participants’ characteristics and dependent variables among groups (*n* = 70).

Variables	Category	VRE (*n* = 23)	IBE (*n* = 24)	Control (*n* = 23)	X² or F *	*p-*Value
		M (SD) or *n* (%)	M (SD) or *n* (%)	M (SD) or *n* (%)		
Age (years)		47.74 ± 5.50	49.00 ± 6.77	48.26 ± 7.56	0.213	0.809
Marital status	Married	20 (87.0)	18 (75.0)	17 (73.9)		
	Single	2 (8.7)	5 (20.8)	4 (17.4)		
	Divorced	0	0	1 (4.3)		
	Widowed	1 (4.3)	1 (4.2)	1 (4.3)	3.515	0.742
Education	Less than high school	7 (30.4)	9 (37.5)	4 (17.4)		
	More than college	16 (69.6)	15 (62.5)	19 (82.6)	2.385	0.303
Employment	Yes	17 (73.9)	19 (79.2)	15 (65.2)		
	No	6 (26.1)	5 (20.8)	8 (34.8)	1.175	0.556
Alcohol consumption	Yes	12 (52.2)	14 (58.3)	14 (60.9)		
	No	11 (47.8)	10 (41.7)	9 (39.1)	0.376	0.829
BMI		26.04 ± 2.25	25.07 ± 2.01	25.39 ± 1.67	1.449	0.242
Depression		5.48 ± 4.50	5.92 ± 3.22	6.00 ± 2.84	0.141	0.869
Exercise fun (NRS)		17.57 ± 8.78	21.42 ± 8.21	16.04 ± 9.55	2.309	0.107
Exercise immersion		30.00 ± 7.20	32.54 ± 8.61	28.57 ± 8.41	1.454	0.241

VRE: Virtual Reality Exercise Group. IBE: Indoor Bicycle Exercise Group. Control: Control Group. BMI (Body Mass Index). * ANOVA (Analysis of Variance).

**Table 2 ijerph-20-00900-t002:** Comparison of BMI, depression, exercise fun, and immersion among groups (*n* = 70).

Variables		VRE (*n* = 23)	IBE (*n* = 24)	Control (*n* = 23)	F * *(p)*	F ** (*p*)
BMI(kg/m^2^)	Baseline	26.04 ± 2.25	25.07 ± 2.00	25.39 ± 1.67	1.449 (0.242)	Time
						109.320 (*p* < 0.001)
	After 4 weeks	25.35 ± 2.09 ^a^	24.86 ± 1.97 ^b^	25.40 ± 1.07 ^c^	30.541 (<0.001)	G*T
						59.491 (*p* < 0.001)
	After 8 weeks	24.65 ± 1.96 ^a^	24.70 ± 1.81 ^b^	25.43 ± 1.78 ^c^	112.527 (<0.001)	Group
						0.548 (*p* = 0.581)
Depression	Baseline	5.48 ± 4.50	5.92 ± 3.22	6.00 ± 2.84	0.141 (0.869)	Time
						16.371 (*p* < 0.001)
	After 4 weeks	3.05 ± 1.89 ^a^	5.54 ± 3.51 ^b^	5.91 ± 3.84 ^b^	7.804 (0.001)	G*T
						3.462 (*p* = 0.010)
	After 8 weeks	2.48 ± 1.47 ^a^	4.17 ± 2.71 ^b^	5.30 ± 3.13 ^b^	9.502 (<0.001)	Group
						3.555 (*p* = 0.034)
Exercise fun	Baseline	17.57 ± 8.78	21.42 ± 8.21	16.04 ± 9.55	2.309 (0.107)	Time
						8.571 (*p* < 0.001)
	After 4 weeks	29.61 ± 5.47 ^a^	18.88 ± 6.65 ^b^	15.35 ± 10.28 ^b^	33.731 (<0.001)	G*T
						12.373 (*p* < 0.001)
	After 8 weeks	32.83 ± 4.61 ^a^	18.92 ± 7.10 ^b^	16.00 ± 10.50 ^b^	40.554 (<0.001)	Group
						14.052 (*p* < 0.001)
Immersion	Baseline	30.00 ± 7.20	32.54 ± 8.61	28.57 ± 8.42	1.454 (0.241)	Time
						7.163 (*p* = 0.002)
	After 4 weeks	42.91 ± 5.49 ^a^	29.38 ± 7.57 ^b^	26.87 ± 9.64 ^b^	39.187 (<0.001)	G*T
						14.629 (*p* < 0.001)
	After 8 weeks	45.91 ± 5.95 ^a^	29.13 ± 6.84 ^b^	27.65 ± 8.90 ^b^	52.346 (<0.001)	Group
						21.219 (*p* < 0.001)

VRE: Virtual Reality Exercise Group. IBE: Indoor Bicycle Exercise Group. Control: Control Group. BMI (Body Mass Index). * ANCOVA (Analysis of Covariance). ** Repeated Measures Analysis of Variance. ^a,b,c^ (Means for each group with different superscript (a, b, c) indicate a significant difference by Bonferroni (*p* < 0.05)).

## Data Availability

Not applicable.

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
