# Peer review of "Virtual Reality Exercise Program Effects on Body Mass Index, Depression, Exercise Fun and Exercise Immersion in Overweight Middle-Aged Women: A Randomized Controlled Trial"

_ijerph, 2023, doi:10.3390/ijerph20020900_

Round 1
Reviewer 1 Report
Virtual Reality Exercise Program Effects on Body Mass Index, Depression, Exercise Fun and Exercise Immersion in Overweight Middle-aged Women: A Randomized Controlled Trial.
This paper contributes to the field of prevention and treatment of a worldwide health problem such as obesity. The authors presented quite a comprehensive introduction, and even when the objective of the study could be inferred, it should be clearly stated: to compare the effectiveness of a virtual reality exercise (VRE) program with the standard indoor bicycle exercise (IBE) and a control group. The methodology used by the authors is adequate and they explain it clearly, so their study could be replicated. Their sample included women with mild depression but the way they presented the sample makes it confuse to track the presence/absence of this condition. Among the main contribution derived of their results is the evidence that points out to virtual reality as a tool generating new fun and value through the immersion of participants in situations they cannot experience in the real world.
My suggestions to help strengthened your paper (in order of appearance):
1) In the introduction, state clearly the objective of this study.
2) In the materials and method section (line 108), you provide information about the City and University using initial letters. It is important to present such information in an open way to give more credibility to your study, so unless there is a compelling reason to keep private the place where the study was conducted (which in that case would be important to explain), I suggest to include the names of the City and University where the study took place.
3) In the materials and method section (line 116), you state that participants with psychiatric disease were excluded. Here I find a major inconsistence because latter, in the discussion section (lines 301-302), you say the average score of the PHQ-9 at baseline, suggesting mild depression in the participants. I believe it is nor your intention but it seems the presence of depression is disregarded in your article. Major Depression is not only a severe worldwide public health problem itself, but it is also a highly prevalent comorbidity with obesity. I suggest you to eliminate the psychiatric exclusion criteria and state at the description of your sample that you included women with mild depression screened with the PHQ-9, otherwise it would appear as a neglected screening. It would be desirable to include this information (women with mild depression) in the title as well; doing that would increase the accuracy of your tittle, and enlarge the scope of your paper into people interested in affective disorders.
4) From what I pointed our above, it is necessary to include data about the comorbidity between obesity and depression in the introduction section. It would be desirable to briefly discuss it at the discussion section.
5) In the material and method section, t is needed to clarify if the experimental maneuver (exercise) was performed at participant´s home, and if was so, how the investigators monitored the subject´s progress throughout the study to make sure they completed all the sessions as they were projected.
6) It would be desirable to briefly discuss the habituation people may experience to virtual reality environments and the effect that could have in long term outcomes of programs using this technology.
I hope these comments may be helpful and will assist you to further strengthening this manuscript.
Reviewer 2 Report
First of all, congratulations for the research work done, I will now mention a number of changes and recommendations in order to obtain clearer and more accurate information on your results.
1. In the introduction section, it would be interesting to include a hypothesis after the objective of the study.
2. Please use also the CONSORT and TidIER or CERT guideline and the checklist for reporting the intervention.
3. Baseline differences: This does not make sense in an RCT, since every baseline difference is by chance because of randomization https://pubmed.ncbi.nlm.nih.gov/7997705/
4. Indicate the reference values of the effect size in the statistical analysis section.
5. Since the abbreviation partial η² is in the material and methods section, it should be used in the results section.
6. In the discussion, the possible causes of obesity in adults could be discussed in more detail.
7. One of your conclusions I recommend including the limitation on the possible bias of not keeping a nutritional diary in order to control for this important variable in the body mass index.
Author Response
On behalf of my co-author, I would like to thank you for the review of our manuscript and for the opportunity to resubmit a revised version. We appreciate the effort and constructive suggestions of the reviewers and have highlighted the revised points in bold accordingly.
Reviewer(s)' Comments and revised version.
Reviewer: 1
Thank you very much for the article review.
Comments)
- In the introduction, state clearly the objective of this study.
Revised> Thank you for pointing this out. We have corrected the sentence in the Introduction following your comments.
In this study, the effect of the exercise intervention method applying virtual reality on body mass index (BMI), depression, exercise fun, and exercise immersion in middle-aged women was investigated.
- In the materials and method section (line 108), you provide information about the City and University using initial letters. It is important to present such information in an open way to give more credibility to your study, so unless there is a compelling reason to keep private the place where the study was conducted (which in that case would be important to explain), I suggest to include the names of the City and University where the study took place.
Revised> Thank you for this valuable suggestion. We have revised the sentence in the Materials and Methods section according to your comments.
The data collection of this study was conducted from February to May 2021 for overweight middle-aged women living in Dae-jeon City through a recruitment notice at Eulji University Hospital.
3) In the materials and method section (line 116), you state that participants with psychiatric disease were excluded. Here I find a major inconsistence because latter, in the discussion section (lines 301-302), you say the average score of the PHQ-9 at baseline, suggesting mild depression in the participants. I believe it is nor your intention but it seems the presence of depression is disregarded in your article. Major Depression is not only a severe worldwide public health problem itself, but it is also a highly prevalent comorbidity with obesity. I suggest you to eliminate the psychiatric exclusion criteria and state at the description of your sample that you included women with mild depression screened with the PHQ-9, otherwise it would appear as a neglected screening. It would be desirable to include this information (women with mild depression) in the title as well; doing that would increase the accuracy of your tittle, and enlarge the scope of your paper into people interested in affective disorders
Revised> Thank you for pointing this out. Following your comments, we have deleted the psychiatric exclusion criterion and modified it to include women with mild depression as tested by the PHQ-9.
The study population included women aged 40–65 years with a BMI of ≥23 kg/m2, those with mild depressive symptoms with a PHQ-9 score of 5–9, those who had not participated in an indoor bicycle exercise program for the past 6 months, those without visual or auditory impairment, those who could use a smartphone, and those who agreed to participate in the study.
4) From what I pointed our above, it is necessary to include data about the comorbidity between obesity and depression in the introduction section. It would be desirable to briefly discuss it at the discussion section.
Revised> Thank you for this suggestion. Per your comments, we have included information about the comorbidity between obesity and depression in the Introduction section as follows:
-In a prospective cohort study of women aged ≥50 years, obesity in middle-aged women was confirmed to affect the long-term risk of developing depression [7]. Mulugeta, A., Zhou, A., Power, C. et al. Obesity and depressive symptoms in mid-life: a population-based cohort study. BMC Psychiatry 18, 297 (2018). https://doi.org/10.1186/s12888-018-1877-6
5) In the material and method section, t is needed to clarify if the experimental maneuver (exercise) was performed at participant´s home, and if was so, how the investigators monitored the subject´s progress throughout the study to make sure they completed all the sessions as they were projected.
Revised> Thank you for this valuable suggestion. According to your comments. We have included the appropriate information in the Material and Method section as follows:
The 8-week exercise application process for the virtual reality and indoor bicycle exercises is as follows: during the RFit application-linked exercise, a stamp was placed on the VRfit record table, and participants were instructed to send it to the researcher at the endpoint after 8 weeks of exercise. Indoor bicycle exercise enabled the transmission of the indoor bicycles’ exercise records as pictures.
6) It would be desirable to briefly discuss the habituation people may experience to virtual reality environments and the effect that could have in long term outcomes of programs using this technology.
Revised> Thank you for this valuable suggestion. We have included the following statement in the Discussion section per your comments:
The best advantage of the virtual reality exercise program is that the external environment has few limitations and can enhance the user's sense of immersion. In addition, since the approach is useful, it has a high possibility of generalization and can induce interest and motivation better than monotonous exercise programs, enabling continuous exercise.
Reviewer: 2
Thank you very much for the review of our article.
Comments)
- In the introduction section, it would be interesting to include a hypothesis after the objective of the study.
Revised> Thank you for this valuable suggestion. Following your comment, we have included the hypotheses after the study's objective in the Introduction section as follows:
Hypothesis 1: There will be a difference in the BMI of the virtual reality exercise, indoor bicycle exercise, or control groups to which the virtual reality exercise program is applied.
Hypothesis 2: There will be a difference in depression between the virtual reality exercise, indoor bicycle exercise, or control groups to which the virtual reality exercise program is applied.
Hypothesis 3: There will be a difference in the exercise fun between the virtual reality exercise, indoor bicycle exercise, or control groups to which the virtual reality exercise program is applied.
Hypothesis 4: There will be differences in exercise immersion among the virtual reality exercise, indoor bicycle exercise, or control groups to which the virtual reality exercise program is applied.
- Please use also the CONSORT and TidIER or CERT guideline and the checklist for reporting the intervention.
Revised>Thank you for this valuable comment. We have included the CONSORT guideline per your comment.
- Baseline differences: This does not make sense in an RCT, since every baseline difference is by chance because of randomization https://pubmed.ncbi.nlm.nih.gov/7997705/
Revised> Thank you for pointing this out. According to your comment, we have revised the relevant part of the statistical analysis section.
- Indicate the reference values of the effect size in the statistical analysis section.
Revised> Thank you for this suggestion. Following the comment, we have revised the relevant part of the statistical analysis section.
If partial η2 is .01, .06, and ≥.14, the effect size is small, medium, or large, respectively. Therefore, the closer the value of partial η2 is to 1, the larger the average difference between groups and the smaller the error.
- Since the abbreviation partial η² is in the material and methods section, it should be used in the results section6. In the discussion, the possible causes of obesity in adults could be discussed in more detail.
Revised> Thank you for this suggestion. Following your comments, we have revised the relevant part of the statistical analysis section.
Cohen, J. (1973). Eta-Squared and Partial Eta-Squared in Fixed Factor Anova Designs. Educational and Psychological Measurement, 33(1), 107–112. https://doi.org/10.1177/001316447303300111
- In the discussion, the possible causes of obesity in adults could be discussed in more detail.
Revised> Thank you for this suggestion. We have revised the relevant part of the Discussion section according to your comment as follows:
Obesity is not only recognized as a disease, but several studies have confirmed that it is an important cause of chronic disease. The number of obese people, including the prevalence of obesity-related chronic diseases, has been gradually increasing recently in many countries worldwide, including Korea; therefore, the problem of obesity is emerging. For this reason, it is time to establish a systematic management plan in Korea.
- One of your conclusions I recommend including the limitation on the possible bias of not keeping a nutritional diary in order to control for this important variable in the body mass index.
Revised> Thank you for this valuable suggestion. Following your comment, we have revised the part of the Conclusion section and included your suggestion as follows:
Therefore, to control the variables of BMI, it is necessary to investigate more objectively in future studies, including nutritional records.
